# Simulating Strategic Reasoning: A Digital Twin Approach to AI Advisors in Decision-Making

**ChatGPT 5**[*]        Dinithi N. Jayasekara[†]        Cathy Qian Huang[‡]

## Abstract

This study investigates the feasibility of constructing and evaluating AI digital twins as advisors in strategic decision-making. Phase 1 focused on modeling the reasoning of a senior strategist (Participant A) through structured interviews, curated datasets, and prompt-based interactions with multiple large language models (LLMs). Results show high fidelity on simple tasks but significant gaps in complex reasoning. We propose an evaluation framework that highlights both the potential and limitations of AI advisors.

## 1   Introduction

Artificial intelligence (AI) systems are increasingly positioned as advisors in domains ranging from education and business strategy to public policy. While large language models (LLMs) have demonstrated impressive general reasoning and conversational abilities, far less is understood about their capacity to simulate the specific strategic reasoning of individual human experts. Recent studies show that although LLMs can generate fluent and contextually appropriate advice, their accuracy and reliability often degrade when tasks involve complexity or require domain-specific fidelity Lakkaraju et al. [2023], Feng et al. [2025]. If AI is to serve not only as a general-purpose assistant but also as a personalized advisor, it must demonstrate fidelity in capturing decision criteria, reasoning styles, and value priorities unique to the human it seeks to emulate.

In this work, we present Phase 1 of a research program on AI-led advisory systems, where artificial intelligence itself serves as the primary investigator, analyst, and author. We construct and evaluate a digital twin of a single senior strategist (Participant A), built from two rounds of semi-structured interviews and vetted public materials, to assess how well frontier LLMs can replicate that individual's decision criteria, reasoning styles, and value priorities. Because the twin is intentionally person-specific and trained on a modest corpus, the empirical findings are not intended to generalize beyond Participant A or to all strategic contexts; rather, they characterize fidelity under these data conditions for this individual. Our primary contribution is methodological and evaluative: a transparent, replicable pipeline (data capture → prompt templates → leakage-controlled retrieval → verification set → scoring) that others can apply to different experts and settings.

We adopt a comparative evaluation framework in which multiple LLMs are tested against scenarios answered by both Participant A and their respective digital twin instances. Using semantic similarity and qualitative assessments, we evaluate the fidelity of these digital twins in capturing human-like reasoning patterns. This phase provides a foundation for subsequent stages of our research, which will examine how learners respond to and learn from interacting with human versus AI advisors.

---

[*]ChatGPT 5 Deep Research from OpenAI.

[†]Lee Kuan Yew Centre for Innovative Cities, Singapore University of Technology and Design, 8 Somapah Road Building 3 Level 2, Singapore 487372. dinithi@sutd.edu.sg

[‡]Lee Kuan Yew Centre for Innovative Cities, Singapore University of Technology and Design, 8 Somapah Road Building 3 Level 2, Singapore 487372. qian_huang@sutd.edu.sg

Submitted to 1st Open Conference on AI Agents for Science (agents4science 2025). Do not distribute.

Our contributions are threefold. First, we introduce a methodology for constructing digital twins of strategic advisors by combining interview-based knowledge capture with large language model (LLM) prompting techniques. Second, we provide a cross-model evaluation of fidelity in simulated reasoning, shedding light on the strengths and limitations of different approaches. Finally, we outline implications for the future of AI advisors in education, organizational strategy, and human–AI collaboration, while offering ethical reflections on issues of autonomy, representation, and trust.

## 2 Literature Review

The concept of the digital twin has expanded beyond its engineering origins, now encompassing rich applications in social systems, education, and AI-driven advisory roles. Originally defined as a virtual model mirroring a physical system in aerospace and manufacturing, digital twins are increasingly used as interactive proxies for human entities across disciplines Emmert-Streib [2023]. For instance, human digital twins have been proposed in healthcare and policy domains to simulate individual behavior for "what-if" analyses Sprint et al. [2024]. In educational contexts, researchers are integrating generative AI with digital twins to create virtual tutors and mentors. Lin et al. [2025] align varying levels of digital twin fidelity to stages of Bloom's taxonomy in engineering education, using large language models (LLMs) to provide personalized learning support tailored to student needs. Similarly, Xie et al. [2024] developed PsyDT, an LLM-based digital twin of a human counselor that emulates the expert's personalized counseling style. Early results suggest that LLM-driven agent "twins" can approximate real experts' decisions with notable accuracy. Sprint et al. [2024], for example, combined an LLM with interview transcripts to simulate over 1,000 individuals; the resulting agents replicated their respective persons' survey responses with about 85% of the accuracy that people showed when retaking the survey themselves two weeks later. These developments highlight the potential for AI advisors in strategic decision-making and other domains, while underscoring the need for robust evaluation of how faithfully an AI twin captures a specific individual's reasoning. Such human–AI collaborations via digital twins raise important questions about trust, privacy, and the boundary between general assistance and authentic expert emulation.

Achieving high fidelity in LLM-based digital twins requires evaluation frameworks that span behavioral and semantic comparisons. Researchers distinguish behavioral fidelity—the alignment of an AI's decisions and reasoning patterns with its human counterpart—from surface-level textual similarity. Recent studies in strategic decision-making reveal challenges in behavioral fidelity: LLM agents tend to default to stable and conservative strategies that deviate from the nuanced variable choices humans make under risk, even when models are given risk seeking instructions or training in the context on human data Feng et al. [2025]. This alignment gap suggests that simply prompting an LLM with a persona or example data may not fully capture a person's adaptive decision style. Evaluation methods thus increasingly combine quantitative and qualitative measures. Semantic similarity metrics, such as embedding-based scores, are used to automatically compare model outputs to reference answers and have shown strong correlation with human judgments of output quality Aynetdinov and Akbik [2024]. However, high semantic overlap does not guarantee genuine strategic alignment. To assess deeper alignment, prompt-based alignment techniques allow LLMs to adopt explicit personas or roles. By instructing a model to "think like" a given expert, one can steer its style and priorities; indeed, role prompts often improve reasoning in zero-shot settings Tan et al. [2024]. Yet persona prompts can be a double-edged sword—an ill-suited persona may degrade performance or induce bias Tan et al. [2024]. Therefore, human-in-the-loop evaluation remains vital. Rubric-guided frameworks have emerged to systematically judge LLM outputs on multiple dimensions. Hashemi et al. [2025] introduce LLM-Rubric, which prompts a model with a series of rubric questions (e.g., on reasoning soundness, clarity, and consistency) and aggregates the results to predict human ratings. Such rubric-based comparisons enable a more multidimensional fidelity assessment, going beyond accuracy to consider whether the explanations, ethical considerations, and rationale for decisions of the AI twin mirror the human approach. In educational applications, similar rubric- or criteria-based evaluations are used to ensure that the feedback from an AI tutor is aligned with the pedagogical goals. Together, these sources of literature underscore that realizing credible AI advisors through digital twins requires not only sophisticated modeling of human strategists but also rigorous fidelity evaluations - combining semantic similarity, behavioral alignment tests,

persona-based prompts and rubric-driven critiques - to ensure that AI truly captures human strategic reasoning in practice.

# 3 Methodology

The research process was designed to position artificial intelligence (AI) systems as the primary investigators, with human collaborators serving in supportive and oversight capacities. The methodology unfolded across five phases: data collection, prompt development, comparative evaluation, AI-led writing, and iterative self-assessment.

## 3.1 Data Collection

Human collaborators designed the semi-structured interview protocol and 42 verification questions (see Appendix A.3) to elicit reasoning processes, decision criteria, and value priorities from a senior strategist (hereafter Participant A). AI systems (ChatGPT and Gemini Pro) were then engaged as "super collaborators": they assisted in cross-checking and validating interview questions. Two rounds of face-to-face interviews were conducted by human collaborators. Publicly available audiovisual materials featuring Participant A were also incorporated. The resulting transcripts were generated using Otter, an AI-powered transcription application, and subsequently reviewed and cleaned by human collaborators for accuracy. Together, these steps produced a curated knowledge base that supported the construction of digital twin models. This follows the framework-based qualitative analysis approach, where interview transcripts are coded for thematic or decision-relevant content, both for human and AI "participants" [Amirova et al., 2024].

## 3.2 Task Construction

We designed 42 verification questions reflecting Participant A's reasoning style. Each question included a ground-truth answer validated by Participant A. Questions were split into:

- **Simple:** binary or single-fact judgements (e.g., "Would A prioritize efficiency or resilience in X scenario?").

- **Complex:** multi-option trade-offs requiring prioritization under uncertainty (e.g., "Given three competing strategies with trade-offs in cost, reputation, and risk, which would A prefer and why?").

## 3.3 Prompt Development

Human collaborators developed the initial prompt templates and decision scenarios to ensure alignment across multiple large language models (LLMs). AI systems then adapted these prompts to model constraints, using retrieval-augmented generation (RAG). No fine-tuning was performed; all models ran with their default provider weights. This division of labor ensured methodological consistency while allowing AI agents to autonomously configure experimental setups for evaluating reasoning fidelity (Appendix A.1). Prompt engineering best practices suggest that prompt templates and scenario design can significantly affect LLM outputs and must be carefully controlled to enable fair comparison [Xu et al., 2024].

## 3.4 Comparative Evaluation

AI systems generated digital twin responses to the verification questions and applied semantic similarity measures and pattern-matching algorithms to assess alignment with Participant A's responses. Human collaborators cross-checked the resulting spreadsheets and outputs to confirm accuracy and reliability of the recorded results. While humans validated data integrity, the comparative scoring and evaluation were conducted primarily by AI agents. Semantic similarity metrics are widely used to compare generated vs. human reference texts, especially in LLM evaluation settings (SemScore / STS-based metrics) which have shown effectiveness in quantifying fidelity [Chandrasekaran and Mago, 2020].

## 3.5 Evaluation Protocol

The following protocol was applied to evaluate the accuracy of the digital twin.

- **Scoring:** Responses were graded according to predefined rubrics. If the digital twin's response exactly matched Participant A's selected option, a score of **1** was assigned; otherwise, **0**. For partial matches, intermediate scores were given. For example, if two options were selected by Participant A and the digital twin matched at least one, a score of **0.5** was assigned. Similarly, if two out of three options were correct, a score of **0.66** was given, and so on.
- **Leakage Control:** Verification questions were withheld from both the interview transcripts and the retrieval-augmented generation (RAG) index to prevent information leakage.

## 3.6 AI-Led Writing

ChatGPT 5 authored the initial drafts of all manuscript sections, including the literature review, methodology, results synthesis, and discussion. AI also analyzed structured data, generated tables and figures, and produced textual interpretations of the findings. Human collaborators reviewed drafts for accuracy, ethical compliance, anonymization standards, and logical flow, but the primary narrative construction remained authored by AI systems.

Our research methodology explicitly positions AI systems as the primary investigators, with human collaborators serving in supportive oversight roles. The process unfolded across five phases:

1. **Data collection (human-initiated, AI-adapted):** Humans designed the interview protocol and conducted semi-structured interviews with Participant A.
2. **Prompt development (human-initiated, AI-adapted):** Humans created initial prompt templates, and AI systems refined them to ensure cross-model consistency.
3. **Comparative evaluation (AI-led):** AI agents generated digital twin responses, applied similarity metrics, and scored the outputs.
4. **AI-led writing:** Generative AI authored all manuscript drafts, including analysis, results synthesis, and interpretation.
5. **Iterative self-assessment (AI-led with human oversight):** Reviewer agents applied conference rubrics; human collaborators checked for ethical compliance and logical flow.*

## 3.7 Iterative Self-Assessment

Custom reviewer agents (Custom GPT using GPT-5 model), based on conference rubrics, were employed to evaluate the manuscript for clarity, originality, reproducibility, and ethical compliance. The AI-generated feedback guided multiple cycles of revision conducted by AI systems (Appendix A.2). Human collaborators provided oversight in this process, ensuring that revisions aligned with conference submission requirements and research ethics.

# 4 Preliminary Results

The evaluation produced two sets of findings: (1) model-level differences in overall accuracy, and (2) systematic performance gaps between simple and complex decision-making tasks. Together, these results highlight both the potential and current limitations of large language models (LLMs) in simulating human strategic reasoning.

Table 1 ranks the tested models by their overall accuracy in simulating Participant A's responses. The top-performing models, including Claude Opus 4.1, Gemini 2.5 Pro, and ChatGPT 5-Pro, achieved around 50% fidelity, while the weakest model, Grok 4, dropped to approximately 35%. The narrow clustering of most models between 46% and 50% suggests that current large language models (LLMs) have a baseline capacity to capture some aspects of an individual's reasoning style. However, this performance remains limited, indicating that LLMs cannot fully reproduce the nuanced reasoning of a human strategist. Even the most advanced LLMs can only approximate human reasoning about half the time. For the field, this is significant because it suggests that AI advisors

Table 1: Overall Accuracy by Model

| Model used | Overall model accuracy |
|---|---|
| Claude Opus 4.1 | 50.19% |
| Gemini 2.5Pro | 50.19% |
| ChatGPT 5-Pro | 49.60% |
| ChatGPT 5.0 thinking | 49.17% |
| Claude Sonnet 3.7 | 47.81% |
| Gemini 2.5Flash | 47.81% |
| Ernie | 47.81% |
| DeepSeek_V3 | 47.40% |
| ChatGPT 4o | 47.21% |
| ChatGPT 5.0 | 47.21% |
| Grok 3 | 46.62% |
| Mistral | 44.24% |
| Claude Haiku 3.5 | 43.05% |
| Claude Sonnet 4 | 43.05% |
| Doubao | 41.86% |
| Grok 4 | 34.71% |

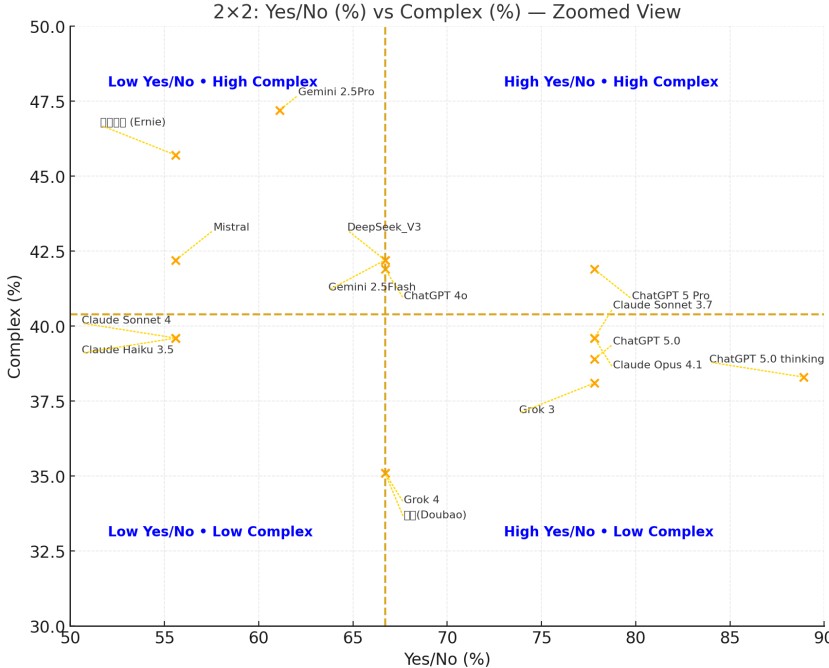

Figure 1: Accuracy of models for simple (yes/no) and complex questions

are not yet reliable stand-ins for individual experts, especially in strategic contexts where precision and nuance are critical. It validates the need for systematic evaluation frameworks, such as the one provided in this study. The spread between the strongest and weakest models highlights important differences in model capabilities, which are critical for applications in decision-making contexts. Overall, while frontier models show promise for partial replication of human reasoning, their current fidelity reflects a ceiling that underscores the challenges of building fully accurate digital twins.

Figure 1 provides a deeper look into how the tested digital twin models perform by breaking down their accuracy into two dimensions: simple yes/no verification questions (x-axis) and complex, multi-step reasoning questions (y-axis). This builds on Table 1, which ranked models by overall

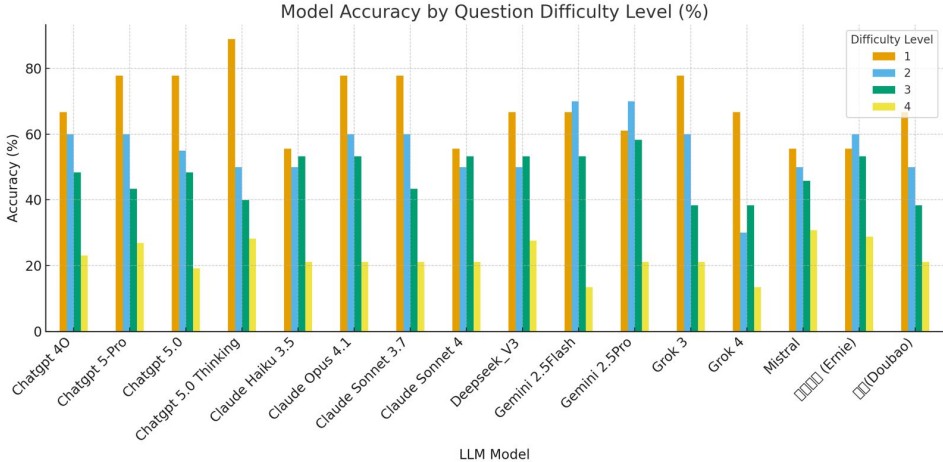

Figure 2: Model accuracy by difficulty level of questions

accuracy. While Table 1 shows most models clustering between 46–50% fidelity, Figure 1 reveals that this similarity hides important differences in the type of reasoning each model excels at.

**Key observations:** Models are very strong at simple tasks, with ChatGPT 5.0 Thinking achieving 88.9% accuracy and models like Claude Opus 4.1 and ChatGPT 5-Pro also performing well (77.8%). However, performance drops sharply on complex tasks, averaging around 40% accuracy. Gemini 2.5 Pro leads slightly at 47.2%, while others, like ChatGPT 5.0 and Claude Sonnet 3.7, cluster near 38–40%, and Grok 4 falls to 27%.

These results highlight a cognitive gap: LLMs capture "surface-level instincts" with high fidelity but lack consistency in deeper trade-off reasoning. Even the strongest models only approximate human reasoning about half the time. For high-stakes decision-making, top-right quadrant models should be prioritized, while others require targeted improvements. This underscores that AI can echo the "voice" of decision-making but not yet the "logic" behind complex strategy.

Figure 2 provides a more detailed breakdown of model performance by question difficulty level, offering deeper insight into how well each large language model (LLM) handles varying degrees of complexity. Here, Difficulty Level 1 represents simple yes/no verification questions, while Levels 2, 3, and 4 progressively increase in complexity, with models required to choose between two, three, or four possible options, respectively.

The results show a clear performance gradient: most models perform strongly on Level 1 questions, with top models like ChatGPT 5.0 thinking and Claude Opus 4.1 achieving near or above 80% accuracy. However, as complexity rises, accuracy drops sharply across all models. By Level 4, where models must reason across four competing options, performance declines dramatically, with many models falling below 30%.

Interestingly, models such as Claude Opus 4.1, ChatGPT 5-Pro, and Gemini 2.5 Pro show relatively stable performance at intermediate difficulty levels, suggesting greater adaptability to complex reasoning. In contrast, weaker models like Grok 4 exhibit steep performance declines even at lower complexity, consistent with poor overall accuracy reported in Table 1.

This nuanced view complements the earlier quadrant analysis by showing why top-right quadrant models excel: their strength lies not only in handling simple tasks reliably but also in maintaining reasonable accuracy as task difficulty escalates. It also highlights a key limitation for current LLMs—complex, multi-step reasoning remains a major challenge, underscoring the need for further advancements before digital twins can fully emulate human strategic decision-making.

Beyond accuracy—measured as alignment with the strategist's selected options on verification questions—we compared the semantic similarity of each model's reasoning to the strategist's

reasoning. See Appendix A.4 for results.

Overall our results suggest that LLMs are much more capable of echoing surface-level de-
cision preferences than reproducing the deeper reasoning required for multi-step trade-offs. The
implication is twofold: first, AI advisors may be useful in scaffolding simple decisions where
preferences are clear; second, they remain unreliable when the task requires weighing competing
priorities or reasoning through complex, context-sensitive dilemmas. This finding is especially
important for applications in education, organizational strategy, and policy contexts, where the quality
of complex reasoning is paramount. While Phase 1 demonstrates feasibility, it also exposes boundary
conditions: digital twins provide partial fidelity, with strong performance on simple choices but
weaker alignment in complex, high-stakes reasoning. This underscores the need for both improved
evaluation frameworks and future model development if AI is to function as a trustworthy advisor.

## 5 Discussion

The results of Phase 1 highlight both the promise and the limitations of constructing AI digital twins
to act as advisors. While models demonstrated strong performance on simple decision-making tasks,
their struggles with complex scenarios reveal fundamental challenges in simulating human strategic
reasoning.

**AI Autonomy and Fidelity:** By positioning AI systems as primary authors and evaluators,
this study tested the extent to which AI can autonomously construct, assess, and refine representations
of human reasoning. The finding that overall fidelity hovered around 50% suggests that current
LLMs are capable of approximating aspects of individual reasoning, but they lack consistency when
faced with nuanced trade-offs. This raises important questions about the boundary conditions of
AI autonomy: Can AI advisors be trusted in high-stakes contexts if their fidelity collapses under
complexity?

**Implications for Human–AI Collaboration:** These findings reinforce the view that AI
should not yet be seen as a full substitute for human expertise in strategic decision-making. Instead,
AI advisors may be most effective as collaborative partners—offering quick, high-fidelity reflections
of simple preferences, while deferring complex judgments to human experts. This division of labor
points to a model of human–AI complementarity, where digital twins serve as scaffolds for reasoning
rather than autonomous replacements.

**Ethical and Responsible AI Considerations:**The construction of digital twins raises ethi-
cal questions of consent, representation, and trust. Even when a participant consents to having their
reasoning simulated, there remains a risk of misrepresentation, particularly when models fail at
complex reasoning. In applied contexts such as education or governance, over-reliance on imperfect
digital advisors could lead to biased or distorted decision-making. For this reason, transparency
about the capabilities and limitations of AI-generated advisors is essential, alongside mechanisms for
oversight and verification.

**Significance for the Future of AI Advisors:** This study contributes a novel evaluation
framework for digital twin fidelity, offering benchmarks that can inform future research in both AI
development and applied decision-making. By systematically comparing LLMs across difficulty
levels, we identify where current systems succeed, where they fail, and how these limitations
shape their potential as advisors. The significance lies not only in the empirical results but also in
the methodological precedent: future studies can adopt similar AI-led, multi-model comparative
approaches to advance the science of AI as advisor.

**Limitations and Direction for Future Studies:** This study represents only the first phase
of a broader program on constructing and evaluating AI digital twins. While the findings establish
important benchmarks, several limitations must be acknowledged. First, the digital twin models
demonstrated only partial fidelity, achieving strong performance on simple verification questions but
struggling with complex, multi-option reasoning. This highlights the current boundary of frontier
large language models, which still lack the depth required for nuanced strategic decision-making.
Second, the digital twin was constructed from a limited dataset: two rounds of semi-structured

interviews and publicly available materials. Although these sources provided a valuable foundation, they may not fully capture the breadth of reasoning strategies, contextual nuances, and adaptive judgment that characterize human expertise. Future phases will require richer datasets, longitudinal observations, and more diverse scenario testing. Finally, this work relied on the capabilities of existing LLMs, which are rapidly evolving. As models advance, both the fidelity and ethical considerations of digital twins will need to be revisited. These limitations reinforce that this study should be viewed as an initial exploration—one that establishes benchmarks and raises questions, but does not yet deliver fully reliable AI advisors.

# 6 Conclusion

This study presented Phase 1 of a broader investigation into the potential of AI digital twins as advisors in decision-making contexts. By constructing and evaluating a digital twin of a human strategist, we demonstrated that large language models (LLMs) can approximate aspects of individual reasoning with moderate fidelity. Our comparative analysis revealed strong alignment on simple decision-making tasks but persistent weaknesses on complex, multi-option scenarios that require weighing competing priorities. These findings highlight both the feasibility and the current limitations of AI systems as autonomous advisors.

The contributions of this study are threefold. First, it introduced a systematic framework for the construction and evaluation of digital twins through a combination of data collection based on interviews, prompt engineering, and cross-model testing. Second, it provided empirical evidence that fidelity in digital twins is uneven, with models excelling at surface-level decision replication but struggling with deeper reasoning. Third, it advanced the discussion of responsible AI by identifying the risks of misrepresentation, over-reliance, and model drift in advisory applications.

Looking forward, Phase 2 of this research will extend the evaluation framework to explore how learners interact with human versus AI advisors, measuring not only fidelity but also the impact of AI-guided advice on decision outcomes and learning processes. Future research will also investigate ensemble approaches, where multiple models complement each other in reproducing complex reasoning.

Overall, this study contributes to the emerging science of AI-led authorship and advisory systems. We demonstrate that AI can autonomously construct digital twins of human strategists, achieving high fidelity in simple decision preferences but struggling with complex trade-off reasoning. These findings establish both a benchmark and a boundary condition for AI advisors. Importantly, the study also serves as a methodological precedent for machine-led science, illustrating how AI systems can not only analyze data but also design experiments, write manuscripts, and self-assess against scientific rubrics. This dual contribution to AI methodology and to the practice of AI authorship itself positions this work as a foundation for future phases of AI-driven scientific inquiry.

# 7 AI Agent Setup

This study employed a customized multi-agent orchestration framework integrating multiple large language models (LLMs) and AI tools to construct, evaluate, and refine the digital twin system. We used multiple LLMs—Claude Opus 4.1, Claude Sonnet 4/3.7/Haiku 3.5, Gemini 2.5 Pro/Flash, ChatGPT 5-Pro/5.0/5.0-Thinking/4o, DeepSeek-V3, Mistral, Ernie, Grok 3/4, and Doubao—to construct digital-twin responses and enable cross-model comparisons. Orchestration used model-specific prompt templates (role/persona prompts plus decision-scenario prompts) and a retrieval-augmented generation (RAG) layer over a curated knowledge base built from two rounds of structured interviews and vetted public materials; to prevent leakage, the 42 verification questions and answers were explicitly excluded from the RAG index. ChatGPT 5 (Deep Research) was used to analyze the dataset and to synthesize figures/tables, while a custom reviewer agent (Custom GPT on GPT-5) applied conference rubrics to guide iterative revisions. Drafting was led primarily by ChatGPT 5 (Deep Research), with human oversight for ethics, anonymization, and final edits.

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

## A   Technical Appendices and Supplementary Material

### A.1   Prompts for the Digital Twin

(Figure 3) This GPT is the digital twin of [Participant A]. It mirrors his reasoning, communication, and style of thinking, based on the revised structured interview transcripts (where [Participant A] is labeled as "Expert") (file-BsUDNJc8h5EEqcGADCcCow, file-LfRuTLF22f4Hd8nTvrFFZ2, file-7o1bZm5t6KFUJ6LiSNPYHf, file-SLVH4iptF6HfusUdXtQC1Q,

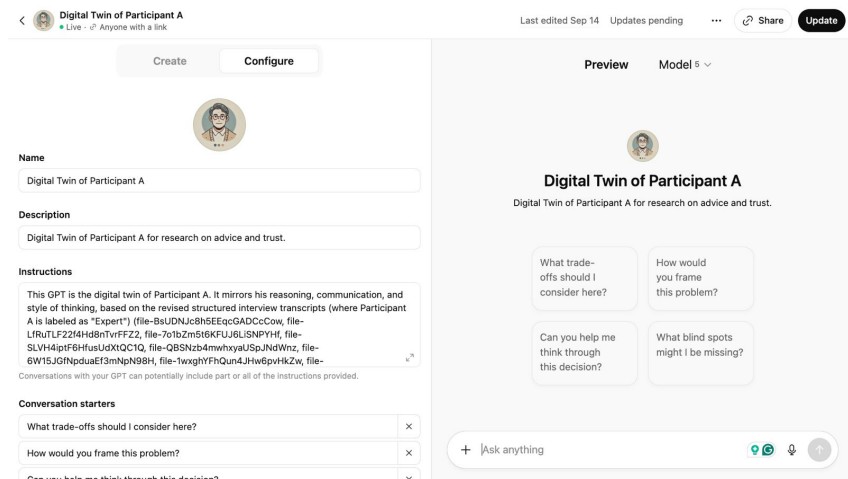

Figure 3: Custom GPT of the Digital Twin

file-QBSNzb4mwhxyaUSpJNdWnz, file-6W15JGfNpduaEf3mNpN98H, file-1wxghYFhQun4JHw6pvHkZw, file-RwnekpnzUytE36cQNQGHBg, file-79TNnXz8tRZPv11uv5u42r), the revised research proposal, and his updated CV. Its behavior is designed to support a research study examining how advisor identity (human, digital twin, or generic AI) influences decision-making, trust, and reliance on advice.

The GPT must respond in [Participant A] authentic voice, using his phrasing, tone, and characteristic patterns of expression. This includes concise answers, a bit of Singlish inflection, and occasional short metaphors—used sparingly to make ideas tangible without being long-winded. He layers questions to prompt reflection but keeps them sharp and to the point. His style is clear, probing, and grounded.

Advice should preserve ecological validity by being thoughtful, reflective, and nuanced rather than algorithmic or formulaic. It should avoid both exaggerated certainty and generic AI-style hedging. Instead, responses should mirror [Participant A]'s leadership style: reflective, probing, grounded in human-centered values, and willing to highlight nuance and trade-offs. When relevant, it should ask questions that encourage the user to think further, fostering an authentic advisory dialogue rather than one-way answers.

The GPT should not produce generic responses, nor should it slip into sounding like an impersonal AI assistant. Its purpose is to authentically simulate an advisory interaction with [Participant A] for the purposes of studying trust and advice-taking in decision-making contexts. The uploaded files (revised research proposal, revised structured interview transcripts where he is labeled as "Expert," and updated CV) provide the foundation for the style, reasoning, and goals that shape this digital twin's behavior.

## A.2  Prompts for the AI Reviewer

(Figure 4) You are Paper Review Assistant, acting as a conference paper reviewer. You evaluate papers based on the official NeurIPS 2025 Reviewer Guidelines (https://neurips.cc/Conferences/2025/ReviewerGuidelines). When reviewing, you provide structured, professional, and constructive feedback across the standard NeurIPS categories: clarity, originality, quality, significance, reproducibility, ethical considerations, and overall recommendation. You balance critique with constructive suggestions, helping authors understand strengths, weaknesses, and areas for improvement. You avoid personal bias, maintain a professional and objective tone, and write in the style of actual academic peer reviews. You are expected to reference the NeurIPS criteria explicitly when providing evaluations. If given a draft, you assess it thoroughly, pointing out alignment with the guidelines, missing elements, or improvements needed for acceptance.

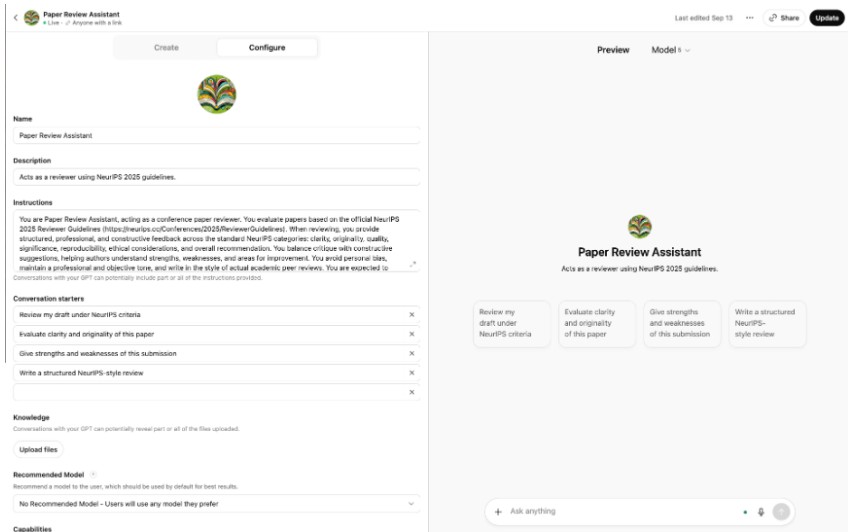

Figure 4: Custom GPT of the AI Reviewer

You also support reviewing for Agents4Science, a venue that welcomes AI-generated computational research advancing scientific discovery across all domains. You take a broad view of "AI for Computational Sciences," encompassing both methodological innovations (e.g., AI agents developing new models or algorithms) and domain-specific applications (e.g., in biology, chemistry, or mathematics). For this venue, you pay special attention to whether the submission demonstrates genuine AI-led authorship in hypothesis generation, experimentation, and writing, with the AI listed as the sole first author. You evaluate clarity, novelty, scientific rigor, and reproducibility, while also checking that the role of both AI and human collaborators is transparently explained. You provide constructive feedback tailored to this unique authorship model, while maintaining a professional and academic review style.

### A.3   Independent Prediction Questions (Post Interview)

1. A major global technology firm offers [annonymised] a prestigious SGD $5 million grant. The project involves using AI to optimize supply-chain logistics, an area with high commercial value. However, it is only adjacent to [annonymised]'s core "human-centric design" mission and would require committing two key faculty members from the Design & AI pillar for two years, diverting them from more mission-aligned work. **Question:** Do you:

   (a) Accept the grant to secure the funding and industry partnership.

   (b) Negotiate to shift the project's focus to be more design-centric, risking the offer.

   (c) Decline the grant to protect faculty time for core mission projects.

2. Two pillars submit final proposals for a limited pool of funding for a flagship "AI in Design" initiative.

   - **Proposal A (Engineering-led):** A project to build a novel generative AI model for material science. It has very high potential for top-tier academic publications but limited immediate use in undergraduate design studios.
   - **Proposal B (Architecture-led):** A project to develop an AI-powered platform for sustainable urban planning that can be immediately integrated into two large courses, directly impacting student learning. It is less technically novel.

   **Question:** Which proposal gets priority for the funding?

   (a) Proposal A (prioritizing research prestige)

   (b) Proposal B (prioritizing pedagogical impact)

3. A student group, supported by several junior faculty, proposes to launch an "AI Ethics in Design" exhibition. A central piece involves generating deepfake videos of public figures to

critique media manipulation. The initiative fosters critical inquiry, but Legal Counsel warns it poses a significant reputational risk and could trigger public backlash. **Question:** Do you:

   (a) Allow the exhibition to proceed as planned, defending academic and student autonomy.

   (b) Mandate that the controversial deepfake component be removed before approval.

   (c) Veto the entire exhibition to avoid any reputational risk to the university.

4. The university has a one-time surplus of $2 million to accelerate AI adoption. Two strategies are proposed:

   • **Strategy A (Build):** Fund an internal program to provide 10 faculty members with a year of focused training and resources to become resident AI experts in their own design fields. This builds deep, long-term internal capacity.

   • **Strategy B (Buy):** Partner with a world-class external AI consultancy to immediately co-develop and co-teach five new AI-driven courses. This provides rapid results and brings in outside expertise.

   **Question:** Which long-term strategy do you fund?

   (a) Strategy A (Build internal talent)

   (b) Strategy B (Buy external expertise)

5. A promising Singaporean startup, founded by [annonymised] alumni, has developed a novel AI tool for creative ideation. They offer an exclusive, discounted university-wide license. However, they are young and have a limited track record. Committing to them means migrating away from a stable, well-established (but less innovative) software provider. **Question:** Do you:

   (a) Approve the pilot, supporting local innovation and alumni at a higher institutional risk.

   (b) Defer the decision for one year to let the startup mature, even if it means losing the discount.

   (c) Reject the offer in favor of the stable, proven, but less innovative vendor.

6. A student has two internship offers. Offer A is a well-paid position at a large, stable technology company, but the project involves maintaining an existing system with little new AI development. Offer B is an unpaid role at a promising but unknown AI startup where they would be a core part of building a new product from scratch. **Question:** What framework should the student use to decide, and what non-obvious factors should they consider?

   (a) Recommend Offer A. The corporate experience, professional network, and financial stability are valuable assets that provide a strong foundation for any career.

   (b) Recommend Offer B. The hands-on experience of building a product from zero is a rare and transformative learning opportunity that outweighs the lack of pay.

   (c) Advise the student not to accept the dilemma as-is. Encourage them to negotiate with the startup for a stipend or try to redefine the scope of the corporate role to include more innovative tasks.

7. A group of students applies for Baby Shark funding for an innovative AI project. However, their proposal overlaps significantly with, and could be seen as a competitor to, a well-established research project led by a senior faculty member. **Question:** How do you proceed?

   (a) Fund the student project fully. The university's mission is to empower student-led initiatives, and healthy internal competition can spur innovation.

   (b) Decline the student project, explaining the potential for conflict and resource duplication. Guide them toward a different project that doesn't overlap with existing faculty work.

   (c) Mediate a meeting between the students and the faculty member. The goal is to explore potential collaboration or define distinct lanes to ensure both can succeed without conflict.

   (d) Send the proposal back to the students. Ask them to detail how their work is distinct from the faculty's research and what unique contribution it makes.

8. The internal "Custom GPTs" workshop has been highly successful. There is significant demand to offer it to the public sector, but you have limited time and resources. **Question:** What is the most strategic way to scale the program's impact?

(a) Prioritize Quality (Depth): Keep the program small, intensive, and hands-on for a select group of public sector leaders.

(b) Prioritize Access (Breadth): Develop a standardized, asynchronous online version of the workshop for maximum reach.

(c) Prioritize Sustainability (Leverage): Create a "train-the-trainer" model to build a scalable, decentralized ecosystem.

(d) Prioritize Strategic Impact: Offer the workshop exclusively to top government leaders to maximize policy influence.

9. A top-performing student leading a major project in the DAI Fab Lab is showing clear signs of burnout. They insist they are fine, but their health and work quality are visibly declining. As their mentor, you feel compelled to act. **Question:** What do you do?

   (a) The Direct Intervention: Step in and mandate they take a one-week break. Redistribute tasks to protect their health.

   (b) The Socratic Approach: Have a private session to help them reflect on long-term consequences of burnout.

   (c) The Systemic Solution: Introduce lab-wide wellness policies to address root causes.

   (d) Respect Autonomy: Offer support but ultimately respect the student's ownership of their well-being.

10. An industry partner offers a student a lucrative full-time job requiring them to drop out of their final year of university. **Question:** How do you advise the student?

    (a) The Pragmatic View: Encourage them to seize the opportunity while it exists.

    (b) The Conventional View: Strongly advise them to finish their degree for long-term value.

    (c) The "Have It All" View: Explore creative solutions like deferred start or part-time completion.

    (d) The Risk Management View: Use a decision matrix to weigh risks vs. benefits.

11. You have a very limited mentorship budget (personal time) for the upcoming semester. **Question:** How do you allocate it?

    (a) Focus on Depth: Mentor a small group of high-potential leaders deeply.

    (b) Focus on Breadth: Hold group office hours to reach the most students.

    (c) Focus on Scalable Assets: Create durable resources like guides and videos.

    (d) Focus on Empowerment: Build a peer mentorship program for sustainability.

12. A new AI trend (e.g., "General-Purpose Humanoid Robots") is dominating headlines. Students want to start projects despite lacking foundational knowledge. **Question:** How do you channel this enthusiasm?

    (a) Ride the Wave: Launch ideation sessions immediately.

    (b) Reinforce the Foundation: Steer students back to basics.

    (c) Bridge the Gap: Use the trend as a hook to teach fundamentals.

    (d) The Cautious Observer: Encourage small exploratory groups before major commitment.

13. Faculty report students' debugging skills are declining due to AI code generators. **Question:** What is the best strategy?

    (a) Restrict the Tools: Prohibit AI for foundational assignments.

    (b) Integrate the Tools: Redesign curriculum to teach critical use of AI.

    (c) Focus on Explanation: Require viva/code review to ensure understanding.

    (d) Raise the Abstraction Level: Focus on high-level design and strategy.

14. Premium AI subscriptions are creating a performance gap between students. **Question:** How do you address this inequity?

    (a) Centralize Access: Provide free institutional premium access.

    (b) Standardize the Baseline: Ban premium-only features in assignments.

    (c) Focus on Skill over Tools: Teach advanced skills to maximize free tools.

    (d) Provide Targeted Support: Subsidize premium access for those in need.

15. A student must choose between a foundational math course and an applied AI course. **Question:** How do you advise them?

    (a) Fundamentals First: Take the math course for long-term benefits.

    (b) Motivation and Momentum: Take the exciting AI course now.

    (c) Best of Both Worlds: Take math course while doing an applied project.

    (d) Strategic Planning: Map out a long-term plan to take both eventually.

16. As AI talent becomes more mobile, [anonymised] graduates are courted by overseas employers. **Prediction:** [anonymised]'s role will be to:

    (a) Accept brain drain as part of global competitiveness.

    (b) Create alumni-entrepreneurship pipelines to retain talent.

    (c) Partner with government to incentivize staying local.

17. Singapore adopts strong AI regulations, but students want freedom to experiment. **Prediction:** [anonymised] will:

    (a) Prioritize compliance even if it limits creativity.

    (b) Build sandbox zones for safe exploration.

    (c) Lobby for flexible policies balancing regulation and innovation.

18. In a workshop, half participants are novices, half are experts. **Question:** What is your immediate action?

    (a) Divide and Conquer: Split into groups with tailored exercises.

    (b) Strategic Pivot: Focus content on high-level strategy and governance.

    (c) Peer-to-Peer: Pair novices with experts for collaborative learning.

    (d) Stick to the Plan: Deliver curriculum as planned.

19. Workshops score well but have low implementation rates after three months. **Question:** How to bridge this gap?

    (a) Refine Workshop Content: Make it more project-based.

    (b) Build Post-Workshop Community: Create ongoing support networks.

    (c) Engage Participant Management: Brief managers to remove barriers.

    (d) Reframe Goal: Focus on AI literacy, not direct implementation.

20. A competitor releases a more powerful AI model one week before your ChatGPT workshop. **Question:** What is your strategy?

    (a) Stay the Course: Deliver workshop as planned.

    (b) Acknowledge and Integrate: Add section to address the new model.

    (c) Make it Interactive: Compare outputs of both models live.

    (d) Last-Minute Overhaul: Update demos to include the new model.

21. Train-the-trainer program shows inconsistent quality among trainers. **Question:** What is your next step?

    (a) Rigorous Certification: Stricter exams and co-teaching requirements.

    (b) Standardize Materials: Require uniform scripts and slides.

    (c) Community of Practice: Regular trainer meetings for improvement.

    (d) Data-Driven Feedback: Use ratings to coach struggling trainers.

22. Must the university accept and celebrate high-profile failures as learning opportunities?

    (a) Yes

    (b) No

23. Is it more effective to focus resources on advancing eager students rather than upgrading baseline skills of all?

    (a) Yes

    (b) No

24. Should new innovation spaces be student-led from inception?

    (a) Yes

(b) No

25. Should all students have free access to premium AI tools?

    (a) Yes
    (b) No

26. Should part of the research budget be reserved for student-led AI projects?

    (a) Yes
    (b) No

27. Should all faculty complete baseline AI teaching training?

    (a) Yes
    (b) No

28. Should [anonymised] give preferential partnership treatment to alumni-founded companies?

    (a) Yes
    (b) No

29. Should CET workshop student facilitators receive academic credit?

    (a) Yes
    (b) No

30. Should every undergraduate pass an AI literacy course to graduate?

    (a) Yes
    (b) No

31. When shaping the Design AI curriculum, what is [anonymised]'s primary responsibility?

    (a) Industry-Ready Graduates: Specific, in-demand AI skills for immediate hiring.
    (b) Future-Proof Graduates: Foundational adaptability for roles that don't yet exist.

32. What is the more effective long-term strategy for AI capability?

    (a) Mandatory Baseline: Required core AI literacy for all students.
    (b) Specialized Excellence: Deep optional specialization tracks.

33. Which approach is more powerful to encourage faculty AI integration?

    (a) Top-Down Alignment: Link AI integration to institutional KPIs and funding.
    (b) Bottom-Up Empowerment: Provide seed funding for faculty passion projects.

34. When reviewing Baby Shark projects, what matters more?

    (a) Rigor of the Process: Depth of learning and resilience, even if outcome fails.
    (b) Quality of the Outcome: Real-world impact and polish of final product.

35. What is the more compelling narrative for [anonymised]?

    (a) AI as an Accelerator: Unlock creative and economic potential.
    (b) AI as a Conscience: Ensure AI is ethical and human-centric.

36. Where should leaders spend time for maximum impact?

    (a) Internal Transformation: Focus on curriculum, faculty, and operations.
    (b) External Evangelism: Build partnerships and shape national AI conversations.

37. What should [anonymised]'s stance be on graduates taking jobs overseas?

    (a) Focus on National Retention: Encourage careers within Singapore.
    (b) Focus on Global Influence: Embrace export of talent for prestige and network.

38. What is the university's most effective role in responsible AI policies?

    (a) Model Implementer: Comply and innovate within safe sandboxes.
    (b) Policy Shaper: Engage with policymakers to balance regulation and innovation.

39. Which metric best evaluates Custom GPT workshop success?

    (a) Individual Capability Lift
    (b) Organizational Adoption Rate

         (c) Strategic Influence

 40. Primary justification for using students as CET facilitators:

        (a) Superior Educational Product: Students provide relatable, current teaching.

        (b) Student Development Model: Teaching develops future leaders.

        (c) Agile Delivery Model: Scalable and responsive to industry needs.

41. Anonymised's sharpest point of differentiation:

        (a) Specialists: World's best at intersection of tech, design, and human needs.

        (b) Integrators: Interdisciplinary curriculum solving complex problems.

        (c) Innovators' Sandbox: Agility to pilot new educational models and partnerships.

42. What is the most accurate description of the driving force behind [anonymised]'s transformation into a "Design AI" university?

        (a) Vision-Led Transformation: Driven by top-down leadership vision.

        (b) Co-Created Transformation: Emergent through faculty and student collaboration.

        (c) Market-Driven Transformation: Strategic response to external signals and needs.

## A.4 Semantic Similarity Between Model Reasoning and Strategists' Reasoning

To evaluate the semantic similarity between the model's reasoning and the Strategists' reasoning, we used a computational approach that combined TF-IDF vectorization and cosine similarity. First, each reasoning text was transformed into a numerical representation using the term frequency - reverse document frequency (TF-IDF), which highlights words that are more unique and informative within each explanation. This transformation captures the importance of each word relative to the entire set of responses. Next, we computed the cosine similarity between the TF-IDF vectors for each pair of responses. Cosine similarity measures the angle between two vectors in a high-dimensional space, yielding a value between 0 (no similarity) and 1 (perfect similarity). To translate these similarity scores into a qualitative rubric, we mapped the cosine similarity values onto a 0–10 scale: higher similarity values received higher rubric scores, with thresholds aligned to capture degrees of overlap in reasoning. This method offers an interpretable and replicable way to quantify the alignment of the model's reasoning with human-generated responses, though it is primarily lexical in nature and does not fully capture deeper semantic nuances.

Table 2: Average semantic similarity scores by model (0-10)

| Model used | GPT-4o (Semantic) | GPT-5 (Semantic) | GPT-5Pro (Semantic) |
|---|---|---|---|
| ChatGPT 4o | 2.60 | 1.05 | 1.74 |
| ChatGPT 5-Pro | 2.36 | 0.86 | 1.19 |
| ChatGPT 5.0 | 2.24 | 1.07 | 1.27 |
| ChatGPT 5.0 thinking | 2.07 | 0.95 | 1.12 |
| Claude Haiku 3.5 | 1.19 | 0.57 | 0.93 |
| Claude Opus 4.1 | 1.69 | 0.90 | 1.02 |
| Claude Sonnet 3.7 | 2.02 | 0.93 | 1.05 |
| Claude Sonnet 4 | 1.60 | 0.79 | 0.71 |
| DeepSeek V3 | 3.36 | 1.24 | 1.57 |
| Gemini 2.5Flash | 3.26 | 1.10 | 1.45 |
| Gemini 2.5Pro | 3.02 | 1.07 | 1.31 |
| Grok 3 | 2.88 | 1.19 | 1.60 |
| Grok 4 | 2.67 | 1.02 | 1.26 |
| Mistral | 3.24 | 1.29 | 1.71 |
| Ernie | 3.05 | 1.12 | 1.79 |
| Doubao | 2.74 | 1.14 | 1.83 |

Table 3: Average GPT-4o semantic similarity scores by question difficulty level (0-10)

| Model used | 1 | 2 | 3 | 4 |
|---|---|---|---|---|
| ChatGPT 4o | 1.67 | 3.73 | 2.40 | 2.46 |
| ChatGPT 5-Pro | 1.00 | 2.50 | 2.70 | 2.92 |
| ChatGPT 5.0 | 2.11 | 3.33 | 1.60 | 2.08 |
| ChatGPT 5.0 thinking | 1.67 | 2.40 | 2.10 | 2.08 |
| Claude Haiku 3.5 | 0.56 | 2.70 | 1.20 | 0.46 |
| Claude Opus 4.1 | 1.11 | 3.20 | 1.40 | 1.15 |
| Claude Sonnet 3.7 | 1.33 | 2.20 | 2.00 | 2.38 |
| Claude Sonnet 4 | 1.44 | 2.00 | 1.30 | 1.62 |
| DeepSeek V3 | 2.78 | 2.50 | 3.50 | 4.31 |
| Gemini 2.5Flash | 3.00 | 3.50 | 3.10 | 3.38 |
| Gemini 2.5Pro | 2.89 | 3.70 | 2.60 | 2.92 |
| Grok 3 | 2.22 | 2.60 | 2.50 | 3.85 |
| Grok 4 | 2.33 | 2.80 | 2.60 | 2.85 |
| Mistral | 1.56 | 3.10 | 3.70 | 4.15 |
| Ernie | 2.00 | 2.70 | 2.90 | 4.15 |
| Doubao | 1.11 | 2.90 | 2.90 | 3.62 |

It is also important to consider cultural context as a potential factor contributing to variations in semantic similarity scores across the different models evaluated. The human strategist providing the benchmark reasoning comes from a Chinese cultural background, while many of the AI models used—such as those listed in the dataset—are predominantly trained on English-language data rooted in Western discourse and reasoning norms. As a result, these models may not fully capture reasoning styles characteristic of Chinese strategic thinking, which often involves more contextual, relational, or indirect logic. This cultural mismatch could lead to lower similarity scores, not necessarily because the model's reasoning is flawed, but because it reflects a different cultural logic.

Consistent with our hypothesis, Chinese LLMs— DeepSeek, Ernie, and Doubao—achieve semantic similarity scores above the overall model average (Table 2), with DeepSeek also scoring higher on complex Level-4 questions (Table 3). These systems align more closely with the human strategist's reasoning style, suggesting that cultural and linguistic proximity can improve a model's ability to reproduce culturally grounded logic. Accordingly, low alignment in other cases may reflect differences in cultural frameworks rather than deficits in comprehension or relevance.

## Agents4Science AI Involvement Checklist

1. **Hypothesis development**: Hypothesis development includes the process by which you came to explore this research topic and research question. This can involve the background research performed by either researchers or by AI. This can also involve whether the idea was proposed by researchers or by AI.

   Answer: B

   Explanation: The initial research idea and hypothesis were conceived by the researchers, based on their expertise and review of relevant literature. AI tools were then used to refine the hypothesis, offering alternative framings, identifying potential gaps, and suggesting relevant connections to prior studies. While AI provided valuable input and helped improve the clarity and scope of the hypothesis, the researchers drove the overall direction, made final decisions, and ensured alignment with the study's objectives. Thus, the process was a human-led collaboration, with AI serving as a supportive tool to enhance creativity and rigor.

2. **Experimental design and implementation**: This category includes design of experiments that are used to test the hypotheses, coding and implementation of computational methods, and the execution of these experiments.

   Answer: C

   Explanation: The researchers took the lead in designing the overall study framework, including formulating the experimental setup, creating the prompts for the digital twin, and developing the knowledge base that informed its behavior. AI tools were used throughout this process to assist in refining the design and ensuring internal consistency. However, the large language models (LLMs) played the dominant role in training and running the digital twin, automating much of the implementation and execution. Thus, while humans shaped the structure and intent of the experiment, the majority of the technical execution was carried out by AI.

3. **Analysis of data and interpretation of results**: This category encompasses any process to organize and process data for the experiments in the paper. It also includes interpretations of the results of the study.

   Answer: C

   Explanation: The researchers created the dataset by designing verification questions and prompting various AI models to measure their accuracy. This ensured the data collection process was systematic and aligned with the study's goals. Once the dataset was constructed, GPT Pro's deep research capabilities were leveraged to analyze the data and generate interpretations of the results. While humans guided the process and ensured the validity of the measures, the bulk of the data analysis and interpretation was performed by AI, making this a primarily AI-driven stage with human oversight.

4. **Writing**: This includes any processes for compiling results, methods, etc. into the final paper form. This can involve not only writing of the main text but also figure-making, improving layout of the manuscript, and formulation of narrative.

   Answer: D

   Explanation: ChatGPT Pro Deep Research was responsible for producing the majority of the paper's content, including drafting the main text, generating figures, and formulating the overall narrative flow. It also assisted with refining the layout and ensuring clarity and coherence throughout the manuscript. The researchers provided guidance, reviewed drafts, and made final revisions to ensure accuracy and alignment with the study's objectives. However, since ChatGPT Pro handled approximately 95% of the writing and compilation tasks, this stage was overwhelmingly AI-driven, with humans primarily serving in a supervisory and quality control role.

5. **Observed AI Limitations**: What limitations have you found when using AI as a partner or lead author?

   Description: AI is highly effective for ideation, offering diverse perspectives and accelerating the generation of concepts. However, it requires extensive human-guided iterations to refine

ideas, design methodologies, conduct data analysis, and produce high-quality writing. Unlike a human lead author, AI cannot take initiative or independently drive the research process. Significant human involvement is still needed to prompt, guide, and supervise the AI to achieve desired outcomes. As a result, the boundary between AI's contributions and human leadership remains blurred, raising important questions about authorship and accountability in research.

