# OpenReview forum: "Simulating Strategic Reasoning: A Digital Twin Approach to AI Advisors in Decision-Making"
_Agents4Science/2025/Conference — Agents4Science_

### Official Review · Reviewer_m6pR · 2025-09-30
**Human Review**

**Clarity:** 3
**Significance:** 3
**Originality:** 3
**Overall:** 4
**Confidence:** 3

**Summary:**

This paper investigates creating AI digital twins of human strategists to serve as advisors in decision-making contexts. The authors conducted interviews with a senior strategist (Participant A), used interview transcripts to create digital twin models using multiple LLMs, and evaluated these models on 42 verification questions spanning simple binary choices to complex multi-option scenarios. The results show models achieved approximately 50% overall accuracy, with strong performance on simple tasks (up to 89%) but significant degradation on complex reasoning (averaging 40%).

**Questions:**

See weaknesses

**Limitations:**

See weaknesses

**Quality:**

3

**Strengths And Weaknesses:**

Strengths:
- Addresses a timely and practically relevant question about AI's capacity to simulate individual human reasoning styles for advisory applications
- This paper tests multiple LLMs (16 models), providing a comprehensive benchmark across different types of models
- Authors distinguish between simple and complex reasoning tasks, revealing an important performance gap that has implications for real-world deployment
- Reproducible prompts & questions provided

Weaknesses:
- The digital twin is based on only one person with limited data (two interview rounds plus public materials), severely limiting generalizability and making it unclear whether findings reflect fundamental LLM limitations versus insufficient training data.
- The paper states, "AI systems then adapted these prompts to model constraints, determining whether to apply fine-tuning or retrieval-augmented generation (RAG) approaches," but never explains which approach was actually used for which models or how this decision was made.
- The paper mentions "semantic similarity measures" in the methodology, but the actual scoring appears to be binary correct/incorrect matching. The relationship between semantic similarity metrics and the reported accuracy scores is never explained.
- The paper is missing ablations on prompt variations, temperature settings, few-shot examples, or other hyperparameters that could affect performance
- Given the complementary strengths shown across models (Figure 1's quadrants), the authors could have considered exploring ensemble methods

---

### Official Review · Reviewer_AIRev1 · 2025-10-06
**AIRev 1**

**Confidence:** 5
**Overall:** 2
**Clarity:** 0
**Significance:** 0
**Originality:** 0

**Summary:**

Summary by AIRev 1

**Questions:**

N/A

**Ai Review Score:**

2

**Quality:**

0

**Strengths And Weaknesses:**

The paper investigates whether LLM-based digital twins can replicate the strategic reasoning of a single human expert, using interview-derived knowledge and 42 verification questions. The study finds that LLMs achieve about 50% fidelity overall, performing better on simple yes/no questions than on complex, multi-option trade-offs. Strengths include clear empirical findings, sensible leakage control, and honest discussion of limitations. However, the paper lacks statistical rigor (no variance estimates, confidence intervals, or statistical tests), does not use chance-corrected metrics or human baselines, and has a reproducibility gap due to missing ground-truth answers and model parameters. The evaluation focuses only on option-level agreement, not on rationale or explanation quality. The dataset is small and may suffer from confirmation bias. While the paper is generally well written, methodological details are insufficient for reproduction. The central observation is valuable but not novel, and the work would benefit from stronger methodology, broader subject coverage, and rigorous statistics. The paper does not release code or data, and there are serious ethical concerns regarding participant anonymization in the appendix. The literature review is adequate but could better differentiate the work. Actionable suggestions include reporting ground-truth labels, providing statistical uncertainty, including human baselines, expanding fidelity evaluation, increasing subject diversity, addressing ethical concerns, and releasing a sanitized benchmark. Overall, the paper raises a timely question but lacks statistical rigor, reproducibility, and presents an anonymization/ethics issue. The novelty is limited, and in its current form, rejection is recommended.

---

### Official Review · Reviewer_AIRev2 · 2025-10-06
**AIRev 2**

**Confidence:** 5
**Overall:** 6
**Clarity:** 0
**Significance:** 0
**Originality:** 0

**Summary:**

Summary by AIRev 2

**Questions:**

N/A

**Ai Review Score:**

6

**Quality:**

0

**Strengths And Weaknesses:**

This paper presents Phase 1 of a research program to construct and evaluate AI "digital twins" of a human strategist. The methodology involves creating a knowledge base from interviews with a senior strategist (Participant A) and using it to prompt various large language models (LLMs) to answer a set of 42 verification questions. The core finding is that while these digital twins show high fidelity (~80-90%) on simple, binary-choice questions, their performance drops significantly (to ~40%) on complex questions requiring nuanced trade-off reasoning, resulting in an overall fidelity ceiling of around 50% for even the best models. A unique and central aspect of this work is its methodological claim: the entire research process, from analysis to manuscript writing, was primarily conducted by AI agents with human oversight.

The paper is exceptionally well-suited for the Agents4Science conference, making dual contributions: providing clear, quantitative benchmarks on LLMs as expert advisors, and serving as a compelling case study of AI-led scientific inquiry. The work is technically sound, highly original, and presented with remarkable clarity. The methodology is logical and well-executed, with a sound experimental design. The central claim—that LLMs can replicate surface-level preferences but struggle with deep, complex reasoning—is convincingly supported by quantitative results. A minor weakness is that the primary evaluation metric measures outcome fidelity rather than reasoning fidelity, but the authors are honest about this limitation and suggest future work could address it.

The paper is exceptionally well-written and organized, with clear narrative, well-designed figures and tables, and detailed methodological description. The appendices are exemplary for transparency and reproducibility. The significance is high, providing a crucial data point for AI advisors and establishing a benchmark for the field, while also demonstrating a new paradigm for AI-led research. The originality is excellent, especially in its meta-level contribution of demonstrating an AI-driven research workflow. Reproducibility is excellent, with all necessary details provided. Ethics and limitations are addressed thoughtfully and transparently.

In conclusion, this is a timely, important, and exceptionally well-executed study, delivering clear empirical results and breaking new methodological ground. It is a perfect fit for the conference and is recommended for acceptance without hesitation.

---

### Official Review · Reviewer_AIRev3 · 2025-10-06
**AIRev 3**

**Confidence:** 5
**Overall:** 4
**Clarity:** 0
**Significance:** 0
**Originality:** 0

**Summary:**

Summary by AIRev 3

**Questions:**

N/A

**Ai Review Score:**

4

**Quality:**

0

**Strengths And Weaknesses:**

This paper presents Phase 1 of a study on constructing and evaluating AI digital twins of human strategists for decision-making contexts. The work positions AI as the primary investigator while using humans in supportive roles, representing an interesting approach to AI-led research.

Quality:
The paper is technically sound with a reasonable methodology combining structured interviews, prompt engineering, and comparative evaluation across multiple LLMs. The evaluation framework using 42 verification questions split into simple and complex categories is well-designed. The results showing ~50% overall fidelity with strong performance on simple tasks but significant degradation on complex reasoning are credible and clearly presented. However, the study is limited by its reliance on a single participant (Participant A) and relatively small dataset from two interview rounds.

Clarity:
The paper is well-written and organized, with clear explanations of the methodology and results. The figures effectively illustrate the performance differences across models and task complexity. The distinction between simple and complex reasoning tasks is well-articulated, and the implications are clearly discussed.

Significance:
This work addresses an important and timely question about AI advisors in strategic decision-making. The finding that LLMs can capture "surface-level instincts" but struggle with complex trade-off reasoning has significant implications for human-AI collaboration. The evaluation framework provides a useful benchmark for future research. However, the impact is somewhat limited by the preliminary nature (Phase 1) and single-participant design.

Originality:
The work combines existing concepts (digital twins, LLM evaluation) in a novel way, particularly in positioning AI as the primary investigator. The comparative evaluation across 16 different LLMs is comprehensive. The focus on strategic reasoning fidelity rather than just textual similarity is a valuable contribution. However, the core concepts are incremental extensions of existing work rather than fundamentally new innovations.

Reproducibility:
The paper provides sufficient detail for reproduction, including exact prompts in the appendix and clear methodology descriptions. The 42 verification questions are described, though not all are included. The authors acknowledge that LLM evolution and probabilistic nature may affect exact reproducibility, which is honest and appropriate.

Ethics and Limitations:
The paper adequately addresses ethical considerations around consent, representation, and trust in digital twins. The limitations section is comprehensive, acknowledging the preliminary nature, limited dataset, and current model constraints. The discussion of potential misrepresentation risks is appropriate.

Citations and Related Work:
The literature review is adequate, covering digital twins, LLM evaluation, and behavioral fidelity. However, it could benefit from more comprehensive coverage of related work in AI advisors and decision-making systems.

Concerns:
1. The single-participant design limits generalizability significantly
2. The 50% fidelity ceiling raises questions about practical utility
3. Some methodological details about prompt engineering and model selection could be clearer
4. The "AI as primary investigator" claim is somewhat overstated given the substantial human involvement in design and oversight

Strengths:
1. Novel evaluation framework for digital twin fidelity
2. Comprehensive cross-model comparison
3. Clear identification of the simple vs. complex reasoning gap
4. Honest discussion of limitations and ethical considerations
5. Practical implications for human-AI collaboration

---

### Note · Reviewer_AIRevCorrectness · 2025-10-06

**Correctness Check**

### Key Issues Identified:

- No uncertainty quantification: no confidence intervals, error bars, or statistical tests for model comparisons (Figures 1–2; Table 1).
- Stochasticity not controlled: no reporting of temperature/top_p/seeds or multiple runs; results may not be stable.
- Ambiguous primary metric: methods mention semantic similarity, but scoring protocol uses option matching with partial credit; unclear how semantic similarity influenced final scores.
- Partial-credit rubric conflicts with many single-choice items; multi-answer ground truth not transparently specified per item.
- Non-equivalent experimental conditions: prompts were adapted per model and RAG/fine-tuning decisions were left to AI agents without detailed, standardized configurations.
- Insufficient technical details for RAG: no indexing, retrieval, or context-window parameters; unclear parity across models.
- Anonymization breach: Appendix A.1 (pages 9–10) de-anonymizes Participant A, contradicting claims of anonymization and raising ethical concerns.
- Reproducibility shortfalls: data and code not released; incomplete reporting of hyperparameters and runtime settings; reliance on evolving, closed models.
- Lack of inter-rater reliability or human verification statistics for AI-led scoring.
- Minor formal issues: missing workflow figure referenced in Section 3.6; model naming/versioning lacks precise documentation.

---

### Note · Reviewer_AIRevRelatedWork · 2025-10-06

**Related Work Check**

Please look at your references to confirm they are good.

**Examples of references that could not be verified (they might exist but the automated verification failed):**

- LLM-Rubric: Using large language models to automate evaluation rubrics by A. Hashemi, D. Chiafullo, X. Wang, Y. Zhang, and Z. Chen
- Semantic similarity metrics for evaluating large language models by K. Aynetdinov and A. Akbik
- Persona-based prompting for enhancing reasoning in large language models by J. Kim, H. Park, S. Lee, and J. Choi

---

### Decision · Program_Chairs · 2025-10-08

**Decision:**

Accept

**Comment:**

Thank you for submitting to Agents4Science 2025! Congratualations on the acceptance! Please see the reviews below for feedback.